# Universal guarantees for decision tree induction via a higher-order splitting criterion

**Guy Blanc**
Stanford University
gblanc@cs.stanford.edu

**Neha Gupta**
Stanford University
nehagupta@cs.stanford.edu

**Jane Lange**
Massachusetts Institute of Technology
jlange@mit.edu

**Li-Yang Tan**
Stanford University
liyang@cs.stanford.edu

## Abstract

We propose a simple extension of *top-down decision tree learning heuristics* such as ID3, C4.5, and CART. Our algorithm achieves provable guarantees for all target functions $f : \{\pm 1\}^n \to \{\pm 1\}$ with respect to the uniform distribution, circumventing impossibility results showing that existing heuristics fare poorly even for simple target functions. The crux of our extension is a new splitting criterion that takes into account the correlations between $f$ and *small subsets* of its attributes. The splitting criteria of existing heuristics (e.g. Gini impurity and information gain), in contrast, are based solely on the correlations between $f$ and its *individual* attributes.

Our algorithm satisfies the following guarantee: for all target functions $f : \{\pm 1\}^n \to \{\pm 1\}$, sizes $s \in \mathbb{N}$, and error parameters $\varepsilon$, it constructs a decision tree of size $s^{\tilde{O}((\log s)^2/\varepsilon^2)}$ that achieves error $\leq O(\mathrm{opt}_s) + \varepsilon$, where $\mathrm{opt}_s$ denotes the error of the optimal size-$s$ decision tree. A key technical notion that drives our analysis is the *noise stability* of $f$, a well-studied smoothness measure.

## 1 Introduction

Decision trees have long been a workhorse of machine learning. Their simple hierarchical structure makes them appealing in terms of interpretability and explanatory power. They are fast to evaluate, with running time and query complexity both scaling linearly with the depth of the tree. Decision trees remain ubiquitous in everyday machine learning applications, and they are at the heart of modern ensemble methods such as random forests [Bre01] and gradient boosted trees [CG16], which achieve state-of-the-art performance in Kaggle and other data science competitions.

**Top-down decision tree learning heuristics.** This focus of our work is on provable guarantees for widely employed and empirically successful *top-down heuristics* for learning decision trees. This includes well-known instantiations such as ID3 [Qui86], C4.5 [Qui93], and CART [Bre17]. These heuristics proceed in a top-down fashion, greedily choosing a "good" attribute to query at the root of the tree, and building the left and right subtrees recursively. In fact, all existing top-down heuristics are *impurity-based*, and can be described in a common framework as follows. Each heuristic is determined by a carefully chosen *impurity function* $\mathscr{G} : [0, 1] \to [0, 1]$, which is restricted to be concave, symmetric around $\frac{1}{2}$, and to satisfy $\mathscr{G}(0) = \mathscr{G}(1) = 0$ and $\mathscr{G}(\frac{1}{2}) = 1$. Given examples $(\boldsymbol{x}, f(\boldsymbol{x}))$ labeled by a target function $f : \{\pm 1\}^n \to \{\pm 1\}$, these heuristics query attribute $x_i$ at the

root of the tree, where $x_i$ approximately maximizes the *purity gain with respect to $\mathscr{G}$*:

$$\text{PurityGain}_{\mathscr{G},f}(x_i) := \mathscr{G}(\text{dist}_{\pm 1}(f)) - \underset{\boldsymbol{b} \sim \{\pm 1\}}{\mathbb{E}}[\mathscr{G}(\text{dist}_{\pm 1}(f_{x_i = \boldsymbol{b}}))],$$

where $f_{x_i = b}$ denotes the subfunction of $f$ with its $i$-attribute set to $b$, and $\text{dist}_{\pm 1}(f) := \min\{\Pr[f(\boldsymbol{x}) = 1], \Pr[f(\boldsymbol{x}) = -1]\}$. For example, ID3 and its successor C4.5 use the binary entropy function $\mathscr{G}(p) = \text{H}(p)$ (the associated purity gain is commonly referred to as "information gain"); CART uses the *Gini criterion* $\mathscr{G}(p) = 4p(1-p)$; Kearns and Mansour, in one of the first papers to study impurity-based heuristics from the perspective of provable guarantees [KM99], proposed and analyzed the impurity function $\mathscr{G}(p) = 2\sqrt{p(1-p)}$. The high-level idea is that $\mathscr{G}(\text{dist}_{\pm 1}(f))$ serves as a proxy for $\text{dist}_{\pm 1}(f)$, the error incurred by approximating $f$ with the best constant-valued classifier. (See the work of Dietterich, Kearns, and Mansour [DKM96] for a detailed discussion and experimental comparison of various impurity functions.)

**An impossibility result for impurity-based heuristics.** Given the popularity and empirical success of these top-down impurity-based heuristics, it is natural to seek broad and rigorous guarantees on their performance. Writing $\mathcal{A}_{\mathscr{G}}$ for the heuristic that uses $\mathscr{G}$ as its impurity function, ideally, one seeks a *universal* guarantee for $\mathcal{A}_{\mathscr{G}}$:

> For *all* target functions $f : \{\pm 1\}^n \to \{\pm 1\}$ and size parameters $s = s(n) \in \mathbb{N}$, the heuristic $\mathcal{A}_{\mathscr{G}}$ builds a decision tree of size not too much larger than $s$ that achieves error close to $\text{opt}_s$, the error of the optimal size-$s$ decision tree for $f$. ($\Diamond$)

Unfortunately, it has long been known (see e.g. [Kea96]) that such a universal guarantee is provably unachievable by *any* impurity-based heuristic. To see this, we consider the uniform distribution over examples and observe that for *all* impurity functions $\mathscr{G}$, the attribute $x_i$ that maximizes $\text{PurityGain}_{\mathscr{G},f}(x_i)$ is the one with the highest correlation with $f$:

$$\text{PurityGain}_{\mathscr{G},f}(x_i) \geq \text{PurityGain}_{\mathscr{G},f}(x_j) \quad \text{iff} \quad \mathbb{E}[f(\boldsymbol{x})\boldsymbol{x}_i]^2 \geq \mathbb{E}[f(\boldsymbol{x})\boldsymbol{x}_j]^2. \qquad (1)$$

(This is a straightforward consequence of the concavity of $\mathscr{G}$; see e.g. Proposition 41 of [BLT20b].[1]) Now consider the target function $f(x) = x_i \oplus x_j$, the parity of two unknown attributes $i, j \in [n]$. On one hand, this function $f$ is computed exactly by a decision tree with 4 leaves (i.e. $\text{opt}_4 = 0$). One the other hand, since $\mathbb{E}[f(\boldsymbol{x})\boldsymbol{x}_k] = 0$ for all $k \in [n]$, *any* impurity-based heuristic $\mathcal{A}_{\mathscr{G}}$—regardless of the choice of $\mathscr{G}$—may build a complete tree of size $\Omega(2^n)$ before achieving any non-trivial error.

**Prior work: provable guarantees for restricted classes of target functions.** In light of such impossibility results, numerous prior works have focused on understanding the performance of impurity-based heuristics when run on *restricted* classes of target functions. Fiat and Pechyony [FP04] studied target functions that are halfspaces and read-once DNF formulas under the uniform distribution; recent work of Brutzkus, Daniely, and Malach [BDM19b] studies read-once DNF formulas under product distributions, and provides theoretical and empirical results on the performance of a variant of ID3 introduced by Kearns and Mansour [KM99]. In a different work [BDM19a], they show that ID3 learns $(\log n)$-juntas in the setting of smoothed analysis. Recent works of Blanc, Lange, and Tan [BLT20b, BLT20a] focus on provable guarantees for monotone target functions. (We discuss other related work in Appendix A.)

## 1.1 Our contributions

We depart from the overall theme of prior works: instead of analyzing the performance of impurity-based heuristics when run on restricted classes of target functions, we propose a simple and efficient extension of these heuristics, and show that it achieves the sought-for universal guarantee ($\Diamond$). In more detail, our main contributions are as follows:

1. *A higher-order splitting criterion.* Recalling the equivalence (1), the splitting criteria of impurity-based heuristics are based solely on the correlations between $f$ and its *individual* attributes. Our splitting criterion is a generalization that takes into account the correlations between $f$ and *small*

*subsets* of its attributes. Instead of querying the attribute $x_i$ that maximizes $\mathbb{E}[f(\boldsymbol{x})\boldsymbol{x}_i]^2$, our algorithm queries the attribute $x_i$ that maximizes:

$$\sum_{\substack{S \ni i \\ |S| \leq d}} (1-\delta)^{|S|}\, \mathbb{E}\left[f(\boldsymbol{x}) \prod_{i \in S} \boldsymbol{x}_i\right]^2, \tag{$\star$}$$

where $\delta \in (0,1)$ is an input parameter of the algorithm (and $d = \Theta(1/\delta)$). This quantity aggregates the correlations between $f$ and subsets of attributes of size up to $d$ containing $i$, where correlations with small subsets contribute more than those with large ones due to the attenuating factor of $(1-\delta)^{|S|}$.

2. *An algorithm with universal guarantees.* We design a simple top-down decision tree learning algorithm based on our new splitting criterion ($\star$), and show that it achieves provable guarantees for all target functions $f : \{\pm 1\}^n \to \{\pm 1\}$, i.e. a universal guarantee of the form ($\diamond$). This circumvents the impossibility results for impurity-based heuristics discussed above.

3. *Decision tree induction as a noise stabilization procedure.* Noise stability is a well-studied smoothness measure of a function $f$ (see e.g. Chapter §2.4 of [O'D14]). Roughly speaking, it measures how often the output of $f$ stays the same when a small amount of noise is applied to the input. Strong connections between noise stability and learnability have long been known [KOS04, KKMS08], and our work further highlights its utility in the specific context of top-down decision tree learning algorithms.

   Indeed, our algorithm can be viewed as a "noise stabilization" procedure. Recalling that each query in the decision tree splits the dataset into two halves, the crux of our analysis is a proof that the average noise stability of both halves is significantly higher than that of the original dataset. Therefore, as our algorithm grows the tree, it can be viewed as refining the corresponding partition of the dataset so that the parts become more and more noise stable. Intuitively, since constant functions are the most noise stable, this is how we show that our algorithm makes good progress towards reducing classification error.

   Our algorithm and its analysis are inspired by "noisy-influence regularity lemmas" from computational complexity theory [OSTW10, Jon16].

## 1.2 Formal statements of our algorithm and main result

**Feature space and distributional assumptions.** We will work in the setting of binary attributes and binary classification, i.e. we focus on the task of learning a target function $f : \{\pm 1\}^n \to \{\pm 1\}$. We will assume that our learning algorithm receives labeled examples $(\boldsymbol{x}, f(\boldsymbol{x}))$ where $\boldsymbol{x} \sim \{\pm 1\}^n$ is uniform random, and the error of a hypothesis $T : \{\pm 1\}^n \to \{\pm 1\}$ is defined to be $\mathrm{error}_f(T) \coloneqq \Pr[f(\boldsymbol{x}) \neq T(\boldsymbol{x})]$ where $\boldsymbol{x} \sim \{\pm 1\}^n$ is uniform random. We write $\mathrm{opt}_s(f)$ to denote $\min\{\mathrm{error}_f(T) : T \text{ is a size-}s \text{ decision tree}\}$; when $f$ is clear from context we simply write $\mathrm{opt}_s$.

**Notation and terminology.** For any decision tree $T$, we say the size of $T$ is the number of leaves in $T$. We refer to a decision tree with unlabeled leaves as a *partial tree*, and write $T^\circ$ to denote such trees. We call any decision tree $T$ obtained from $T^\circ$ by a labeling of its leaves a *completion* of $T^\circ$. Given a partial tree $T^\circ$ and a target function $f : \{\pm 1\}^n \to \{\pm 1\}$, there is a canonical completion of $T^\circ$ that minimizes classification error $\mathrm{error}_f(T)$: label every leaf $\ell$ of $T^\circ$ with $\mathrm{sign}(\mathbb{E}[f_\ell])$, where $f_\ell$ denotes the subfunction of $f$ obtained by restricting its attributes according to the path that leads to $\ell$. We write $T^\circ_f$ to denote this canonical completion, and call it the *$f$-completion of $T^\circ$*. For a leaf $\ell$ of a partial tree $T^\circ$, we write $|\ell|$ to denote its depth within $T^\circ$, the number of attributes queried along the path that leads to $\ell$. We will switch freely between a decision tree $T$ and the function $T : \{\pm 1\}^n \to \{\pm 1\}$ that it represents.

We use **boldface** to denote random variables (e.g. '$\boldsymbol{x}$'), and unless otherwise stated, all probabilities and expectations are with respect to the uniform distribution.

**Fourier analysis of boolean functions.** We will draw on notions and techniques from the Fourier analysis of boolean functions; for an in-depth treatment of this topic, see [O'D14]. Every function $f : \{\pm 1\}^n \to \mathbb{R}$ can be uniquely expressed as a multilinear polynomial via its *Fourier expansion*:

$$f(x) = \sum_{S \subseteq [n]} \widehat{f}(S) \prod_{i \in S} x_i, \quad \text{where } \widehat{f}(S) = \mathbb{E}\left[f(\boldsymbol{x}) \prod_{i \in S} \boldsymbol{x}_i\right]. \tag{2}$$

**Definition 1** ($\delta$-noisy influence). *For a function $f : \{\pm 1\}^n \to \mathbb{R}$ and a coordinate $i \in [n]$, the $\delta$-noisy influence of $i$ on $f$ is the quantity*

$$\mathrm{Inf}_i^{(\delta)}(f) := \sum_{S \ni i} (1 - \delta)^{|S|} \widehat{f}(S)^2.$$

*The $\delta$-noisy $d$-wise influence of $i$ on $f$ is the quantity*

$$\mathrm{Inf}_i^{(\delta, d)}(f) := \sum_{\substack{S \ni i \\ |S| \leq d}} (1 - \delta)^{|S|} \widehat{f}(S)^2.$$

We are now ready to state our algorithm and main result:

---

BUILDSTABILIZINGDT$_f(t, \delta, \varepsilon)$:

Initialize $T^\circ$ to be the empty tree.

while (size($T^\circ$) < t) {
    1. *Score:* For every leaf $\ell$ in $T^\circ$, let $x_i$ denote the attribute with the largest $\delta$-noisy $d$-wise influence on the subfunction $f_\ell$ where $d := \log(1/\varepsilon)/\delta$:

$$\mathrm{Inf}_i^{(\delta, d)}(f_\ell) \geq \mathrm{Inf}_j^{(\delta, d)}(f_\ell) \quad \text{for all } j \in [n].$$

    Assign $\ell$ the score:

$$\mathrm{score}(\ell) := \Pr_{\boldsymbol{x} \sim \{\pm 1\}^n} [\, \boldsymbol{x} \text{ reaches } \ell \,] \cdot \mathrm{Inf}_i^{(\delta, d)}(f_\ell) = 2^{-|\ell|} \cdot \mathrm{Inf}^{(\delta, d)}(f_\ell).$$

    2. *Split:* Let $\ell^\star$ be the leaf with the highest score, and $x_i^\star$ be the associated high noisy-influence attribute. Grow $T^\circ$ by splitting $\ell$ with a query to $x_{i^\star}$.

}
Output the $f$-completion of $T^\circ$.

---

Figure 1: BUILDSTABILIZINGDT has access to uniform random examples $(\boldsymbol{x}, f(\boldsymbol{x}))$ of a target function $f : \{\pm 1\}^n \to \{\pm 1\}$, and outputs a size-$T$ decision tree hypothesis.

**Theorem 1** (Main result). *Let $f : \{\pm 1\}^n \to \{\pm 1\}$ be any target function. For all $s \in \mathbb{N}$, $\varepsilon \in (0, \frac{1}{2})$, and $\delta = \varepsilon/(\log s)$, BUILDSTABILIZINGDT$_f$ runs in time $(ns)^{\tilde{O}((\log s)^2/\varepsilon^2)}$ and returns a tree $T$ of size $s^{\tilde{O}((\log s)^2/\varepsilon^2)}$ satisfying $\mathrm{error}_f(T) \leq O(\mathsf{opt}_s) + \varepsilon$.*

Theorem 1 should be contrasted with the impossibility result discussed in the introduction, which shows that there are very simple target functions for which $\mathsf{opt}_4 = 0$, and yet any impurity-based heuristic may build a complete tree of size $\Omega(2^n)$ (in time $\Omega(2^n)$) before achieving any non-trivial error ($< \frac{1}{2}$). Hence, Theorem 1 shows the increased power of our new splitting criterion.

**Remark 1** (Estimating noisy influence from random labeled examples). For all values of $d, i,$ and $\delta$, the quantity $\mathrm{Inf}_i^{(\delta, d)}(f)$ can be estimated to high accuracy from uniform random examples $(\boldsymbol{x}, f(\boldsymbol{x}))$ in time $n^{O(d)}$ via the identity (2). More generally, $\mathrm{Inf}_i^{(\delta, d)}(f_\ell)$ can be estimated to high accuracy in time $\mathrm{poly}(2^\ell, n^d)$. In our analysis we will assume that these quantities can be computed *exactly* in time $\mathrm{poly}(2^\ell, n^d)$, noting that the approximation errors can be easily incorporated into our analysis via standard methods.

**Other useful definitions and identites from the Fourier analysis of boolean functions.** For $f : \{\pm 1\}^n \to \mathbb{R}$, we have Parseval's identity,

$$\sum_{S \subseteq [n]} \widehat{f}(S)^2 = \mathbb{E}[f(\boldsymbol{x})^2]. \tag{3}$$

In particular, for $f : \{\pm 1\}^n \to \{\pm 1\}$ we have that $\sum_{S \subseteq [n]} \widehat{f}(S)^2 = 1$. The *variance of $f$* is given by the formula

$$\mathrm{Var}(f) := \mathbb{E}[f(\boldsymbol{x})^2] - \mathbb{E}[f(\boldsymbol{x})]^2 = \sum_{S \neq \emptyset} \widehat{f}(S)^2. \tag{4}$$

For $x \in \{\pm 1\}^n$, we write $\tilde{\boldsymbol{x}} \sim_\delta x$ to denote a *$\delta$-noisy copy of $x$*, where $\tilde{\boldsymbol{x}}$ is obtained by rerandomizing each coordinate of $x$ independently with probability $\delta$. (Equivalently, $\tilde{\boldsymbol{x}}$ is obtained by flipping each coordinate of $x$ independently with probability $\frac{\delta}{2}$.) For $f : \{\pm 1\}^n \to \{\pm 1\}$ and $\delta \in (0, 1)$, we define the *$\delta$-smoothed version of $f$* to be $f_\delta : \{\pm 1\}^n \to [-1, 1]$,

$$f_\delta(x) := \mathop{\mathbb{E}}_{\tilde{\boldsymbol{x}} \sim_\delta x}[f(\tilde{\boldsymbol{x}})] = \sum_{S \subseteq [n]} (1 - \delta)^{|S|} \widehat{f}(S) \prod_{i \in S} x_i. \tag{5}$$

For $g : \{\pm 1\}^n \to \mathbb{R}$, we write $g^{\leq d} : \{\pm 1\}^n \to \mathbb{R}$ to denote the degree-$d$ polynomial that one obtains by truncating $g$'s Fourier expansion to degree $d$: $g^{\leq d}(x) = \sum_{S \subseteq [n]_{|S| \leq d}} \widehat{g}(S) \prod_{i \in S} x_i$. The *$i$-th discrete derivative* of $g : \{\pm 1\}^n \to \mathbb{R}$ is the function

$$(D_i g)(x) := \frac{g(x^{i=1}) - g(x^{i=-1})}{2} = \sum_{S \ni i} \widehat{g}(S) \prod_{j \in S \setminus \{i\}} x_j,$$

where $x^{i=b}$ denotes $x$ with its $i$-th coordinate set to $b$.

**Definition 2** (Defining influence in terms of derivatives). *For $g : \{\pm 1\}^n \to \mathbb{R}$ and $i \in [n]$, the influence of $i$ on $g$ is the quantity*

$$\mathrm{Inf}_i(g) := \mathbb{E}[(D_i g)(\boldsymbol{x})^2] = \sum_{S \ni i} \widehat{g}(S)^2.$$

*We define the $\delta$-noisy influence of $i$ on $g$ to be the quantity*

$$\mathrm{Inf}_i^{(\delta)}(g) := (1 - \delta) \mathop{\mathbb{E}}_{\substack{\boldsymbol{x} \sim \{\pm 1\}^n \\ \tilde{\boldsymbol{x}} \sim_\delta \boldsymbol{x}}}[(D_i g)(\boldsymbol{x})(D_i g)(\tilde{\boldsymbol{x}})] = \sum_{S \ni i} (1 - \delta)^{|S|} \widehat{g}(S)^2,$$

*and the $\delta$-noisy $d$-wise influence of $i$ on $g$ to be $\mathrm{Inf}_i^{(\delta, d)}(g) := \mathrm{Inf}_i^{(\delta)}(g^{\leq d})$.*

## 2 Our algorithm as a noise stabilization procedure

As alluded to in the introduction, our algorithm BUILDSTABILIZINGDT can be viewed as a "noise stabilization" procedure. Towards formalizing this, we begin by recalling the definition of noise sensitivity:

**Definition 3** (Noise sensitivity of $f$). *For $f : \{\pm 1\}^n \to \{\pm 1\}$ and $\delta \in (0, 1)$, the noise sensitivity of $f$ at noise rate $\delta$ is the quantity $\mathrm{NS}_\delta(f) := \Pr[f(\boldsymbol{x}) \neq f(\tilde{\boldsymbol{x}})]$, where $\boldsymbol{x} \sim \{\pm 1\}^n$ is uniform random, and $\tilde{\boldsymbol{x}}$ is obtained from $\boldsymbol{x}$ by rerandomizing each of its coordinates independently with probability $\delta$.*[2]

Noise sensitivity has been extensively studied (see e.g. O'Donnell's Ph.D. thesis [O'D03]); we will only need very basic properties of this notion, all of which is covered in Chapter §2.4 of the book [O'D14].

We now introduce the potential function that will facilitate our analysis of BUILDSTABILIZINGDT:

**Definition 4** (Noise sensitivity of $f$ with respect to $T^\circ$). *Let $f : \{\pm 1\}^n \to \{\pm 1\}$ be a function and $T^\circ$ be a partial decision tree. The noise sensitivity of $f$ with respect to the partition induced by $T^\circ$ is the quantity*

$$\mathrm{NS}_\delta(f, T^\circ) := \sum_{\text{leaves } \ell \,\in\, T^\circ} \Pr[\boldsymbol{x} \text{ reaches } \ell] \cdot \mathrm{NS}_\delta(f_\ell) = \sum_{\text{leaves } \ell \,\in\, T^\circ} 2^{-|\ell|} \cdot \mathrm{NS}_\delta(f_\ell).$$

We observe that $\mathrm{NS}_\delta(f, T^\circ)$ where $T^\circ$ is the empty tree is simply $\mathrm{NS}_\delta(f)$, the noise sensitivity of $f$ at noise rate $\delta$. The following fact quantifies the decrease in $\mathrm{NS}_\delta(f, T^\circ)$ when we split a leaf of $T^\circ$:

**Fact 2.1** (Stability gain associated with a split). *Let $f : \{\pm 1\}^n \to \{\pm 1\}$ be a function and $T^\circ$ be a partial decision tree. Fix a leaf $\ell^\star$ of $T^\circ$ and a coordinate $i \in [n]$, and let $T^\circ_{\ell^\star \to x_i}$ denote the extension of $T^\circ$ obtained by splitting $\ell$ with a query to $x_i$. We define*

$$\mathrm{StabilityGain}_f(T^\circ, \ell^\star, i) := \mathrm{NS}_\delta(f, T^\circ) - \mathrm{NS}_\delta(f, T^\circ_{\ell^\star \to x_i}).$$

*Then*

$$\mathrm{StabilityGain}_f(T^\circ, \ell^\star, i) = \tfrac{\delta}{2(1-\delta)} \cdot \mathrm{Inf}_i^{(\delta)}(f_{\ell^\star}).$$

We defer proof of Fact 2.1 to Appendix B.

## 3 Every split makes good progress

Fact 2.1 motivates the following question: given a partial decision tree $T^\circ$ and a target function $f$, is there a leaf $\ell^\star$ of $T^\circ$ and a coordinate $i \in [n]$ such that $\mathrm{Inf}_i^{(\delta)}(f_{\ell^\star})$ is large? To answer this question we draw on a powerful concentration inequality of O'Donnell, Saks, Schramm, and Servedio [OSSS05], which can be viewed as a variant of the Efron–Stein [ES81, Ste86] and Kahn–Kalai–Linial inequalities [KKL88].

**Theorem 2** (Theorem 3.3 of [OSSS05]). *Let $T : \{\pm 1\}^n \to \{\pm 1\}$ be a depth-$k$ decision tree and $\rho : \mathbb{R} \times \mathbb{R} \to \mathbb{R}$ be a semimetric. For all functions $g : \{\pm 1\}^n \to \mathbb{R}$, writing $\boldsymbol{x}, \boldsymbol{x}' \sim \{\pm 1\}^n$ to denote uniform random and independent inputs and $\boldsymbol{x}^{\sim i}$ to denote $\boldsymbol{x}$ with its $i$-th coordinate rerandomized,*

$$|\mathrm{Cov}_\rho(T, g)| \le \mathrm{Def}_k(\rho) \sum_{i=1}^n \lambda_i(T) \cdot \mathbb{E}[\rho(g(\boldsymbol{x}), g(\boldsymbol{x}^{\sim i}))],$$

*where*

$$\mathrm{Cov}_\rho(T, g) := \mathbb{E}[\rho(T(\boldsymbol{x}), g(\boldsymbol{x}'))] - \mathbb{E}[\rho(T(\boldsymbol{x}), g(\boldsymbol{x}))],$$

$$\mathrm{Def}_k(\rho) := \max_{z_0, \dots, z_k \in \mathbb{R}} \left\{ \frac{\rho(z_0, z_k)}{\sum_{j=1}^k \rho(z_{j-1}, z_j)} \right\}, \quad \text{where } \tfrac{0}{0} \text{ is taken to be 1,}$$

$$\lambda_i(T) := \Pr[T \text{ queries } \boldsymbol{x}_i].$$

We apply the OSSS ienquality to deduce the following theorem, which will be a key ingredient in our proof of Theorem 1.

**Theorem 3** (Lower bound on the progress of each split). *Let $f : \{\pm 1\}^n \to \{\pm 1\}$ be a function and $T^\star : \{\pm 1\}^n \to \{\pm 1\}$ be a size-$s$ depth-$k$ decision tree. There is an $i \in [n]$ such that*

$$\mathrm{Inf}_i^{(\delta, d)}(f) \ge \frac{\tfrac{1}{2} \mathrm{Var}(f_\delta) - \left( 2 \mathbb{E}[(T^\star(\boldsymbol{x}) - f_\delta(\boldsymbol{x}))^2] + \tfrac{5}{2}\varepsilon \right)}{k \log s},$$

*where $d = \log(1/\varepsilon)/\delta$.*

We defer the proof of Theorem 3 to Appendix C.

## 4 Proof of Theorem 1

We will show that Theorem 1 follows as a consequence of the following more general result, which gives a performance guarantee for BUILDSTABILIZINGDT that scales according to the noise sensitivity of the target function $f$.

**Theorem 4.** *Let $f : \{\pm 1\}^n \to \{\pm 1\}$ and $\delta, \kappa \in (0, 1)$ be such that $\mathrm{NS}_\delta(f) \le \kappa$. For all $s \in \mathbb{N}$ and $\varepsilon \in (0, \tfrac{1}{2})$, by choosing $t = s^{\tilde{\Theta}((\kappa \log s)/\varepsilon \delta)}$, BUILDSTABILIZINGDT$_f(t, \delta, \varepsilon)$ runs in time $\mathrm{poly}(t, n^{\log(1/\varepsilon)/\delta})$ and returns a tree $T$ of size $t$ satisfying $\mathrm{error}_f(T) \le O(\mathsf{opt}_s + \kappa) + \varepsilon$.*

### 4.1 Proof of Theorem 1 assuming Theorem 4

For $f, g : \{\pm 1\}^n \to \{\pm 1\}$, we write $\mathrm{dist}(f, g)$ to denote $\Pr[f(\boldsymbol{x}) \ne g(\boldsymbol{x})]$, and we say that $f$ and $g$ are *$\alpha$-close* if $\mathrm{dist}(f, g) \le \alpha$. We will need two standard facts from the Fourier analysis of boolean functions, whose proofs are included in Appendix D.

**Fact 4.1** (Noise sensitivity of size-$s$ decision trees). *For all size-$s$ decision trees $T$ and $\delta \in (0,1)$,* $\mathrm{NS}_\delta(T) \leq \delta \log s$.

**Fact 4.2.** *For all $f, T : \{\pm 1\}^n \to \{\pm 1\}$ and $\delta \in (0,1)$, $\mathrm{NS}_\delta(f) \leq \mathrm{NS}_\delta(T) + 2\,\mathrm{dist}(f,T)$.*

With these facts in hand we are ready to prove Theorem 1. Let $T_{\mathsf{opt}}$ be a size-$s$ decision tree that is opt-close to $f$. By Facts 4.1 and 4.2, we have $\mathrm{NS}_\delta(f) \leq \mathrm{NS}_\delta(T_{\mathsf{opt}}) + \mathrm{dist}(f, T_{\mathsf{opt}}) \leq \delta \cdot \log s + 2\,\mathsf{opt}$ for all $\delta \in (0,1)$. Choosing $\delta = \varepsilon / \log s$, we have that $\mathrm{NS}_\delta(f) \leq 2\,\mathsf{opt} + \varepsilon$. Plugging this bound into Theorem 4 yields Theorem 1.

## 4.2 Proof of Theorem 4

Let $T_{\mathsf{opt}}$ be a size-$s$ decision tree that is $\mathsf{opt}_s$-close to $f$. While $T_{\mathsf{opt}}$ may have depth as large as $s$, we can define $T_{\mathsf{opt}}^{\mathrm{trunc}}$ to be $T_{\mathsf{opt}}$ truncated to depth $\log(s/\varepsilon)$, replacing all truncated paths with a fixed but arbitrary value (say 1). We have that $\mathrm{dist}(T_{\mathsf{opt}}, T_{\mathsf{opt}}^{\mathrm{trunc}}) \leq \varepsilon$, and so $\mathrm{dist}(f, T_{\mathsf{opt}}^{\mathrm{trunc}}) \leq \mathsf{opt}_s + \varepsilon$. In words, $f$ is $(\mathsf{opt}_s + \varepsilon)$-close to a decision tree $T_{\mathsf{opt}}^{\mathrm{trunc}}$ of size $s$ and depth $\log(s/\varepsilon)$.

Let $T^\circ$ be a partial decision tree, which we should think of as one that BUILDSTABILIZINGDT has constructed after a certain number of iterations. We define the natural probability distribution over the leaves of $T^\circ$ where each leaf $\ell$ receives weight $2^{-|\ell|}$, and write $\boldsymbol{\ell}$ to denote a random leaf of $T^\circ$ drawn from this distribution.

We consider two cases depending on the value of $\mathbb{E}_{\boldsymbol{\ell}}[\mathrm{Var}((f_{\boldsymbol{\ell}})_\delta)]$; the analyses of these cases are deferred to Appendix E.

**Case 1:** $\mathbb{E}_{\boldsymbol{\ell}}[\mathrm{Var}((f_{\boldsymbol{\ell}})_\delta)] \geq 4\,\mathbb{E}_{\boldsymbol{\ell}}\left[\|(f_{\boldsymbol{\ell}})_\delta - T_{\mathsf{opt}}^{\mathrm{trunc}}\|_2^2\right] + 7\varepsilon$.

In this case, we will show in Appendix E that Theorem 3 (our main result from Section 3) implies the existence of a leaf $\ell^\star \in T^\circ$ such that

$$\mathrm{score}(\ell^\star) = 2^{-|\ell^\star|} \cdot \max_{i \in [n]} \left\{ \mathrm{Inf}_i^{(\delta, d)}(f_{\ell^\star}) \right\} \geq \frac{\varepsilon}{|T^\circ| \log(s/\varepsilon) \log s},$$

where $|T^\circ|$ denotes the size of $T^\circ$.

**Case 2:** $\mathbb{E}_{\boldsymbol{\ell}}[\mathrm{Var}((f_{\boldsymbol{\ell}})_\delta)] < 4\,\mathbb{E}_{\boldsymbol{\ell}}\left[\|(f_{\boldsymbol{\ell}})_\delta - T_{\mathsf{opt}}^{\mathrm{trunc}}\|_2^2\right] + 7\varepsilon$.

In this case, we will show in Appendix E that $\mathrm{error}_f(T_f^\circ) \leq O(\mathsf{opt}_s + \kappa + \varepsilon)$.

With this case split in hand, we are now ready to prove Theorem 4. For $j \in [t]$, we write $T_j^\circ$ to denote the size-$j$ partial decision tree that BUILDSTABILIZINGDT constructs after $j-1$ iterations. Therefore $(T_t^\circ)_f$, the $f$-completion of $T_t^\circ$, is the final size-$t$ decision tree that it returns.

For any $j \leq t$, if $T_j^\circ$ falls into Case 2, we have shown that $\mathrm{error}_f((T_j^\circ)_f) \leq O(\mathsf{opt}_s + \kappa + \varepsilon)$. Since classification error cannot increase with further splits, i.e. $\mathrm{error}_f((T_t^\circ)_f) \leq \mathrm{error}_f((T_j^\circ)_f)$, it suffices to bound the maximum number of times we fall into Case 1. We will use our potential function, the noise sensitivity of $f$ with respect to a partial decision tree $T^\circ$ (Definition 4), to bound this number.

First, we have that $\mathrm{NS}_\delta(f, T_1^\circ) = \mathrm{NS}_\delta(f) \leq \kappa$ (i.e. the potential function starts off being at most $\kappa$). For every $j \geq 1$, if $T_j^\circ$ falls into Case 1, we have that BUILDSTABILIZINGDT splits the leaf $\ell^\star$ with the highest score, which we have shown to be at least

$$\mathrm{score}(\ell^\star) = 2^{-|\ell^\star|} \cdot \max_{i \in [n]} \left\{ \mathrm{Inf}_i^{(\delta, d)}(f_{\ell^\star}) \right\} \geq \frac{\varepsilon}{j \log(s/\varepsilon) \log s}.$$

Equivalently, BUILDSTABILIZINGDT splits $\ell^\star$ with a query to an attribute $x_i$, where

$$\mathrm{Inf}_i^{(\delta, d)}(f_{\ell^\star}) \geq 2^{|\ell^\star|} \cdot \frac{\varepsilon}{j \log(s/\varepsilon) \log s}. \tag{6}$$

Since $\mathrm{Inf}_i^{(\delta)}(f_{\ell^\star}) \geq \mathrm{Inf}_i^{(\delta, d)}(f_{\ell^\star})$ it follows from Fact 2.1 and Equation (6) that

$$\mathrm{NS}_\delta(f, T_{j+1}^\circ) = \mathrm{NS}_\delta(f, T_j^\circ) - \mathrm{StabilityGain}(T_j^\circ, \ell^\star, i),$$

where

$$\mathrm{StabilityGain}(T_j^\circ, \ell^\star, i) \geq \frac{\varepsilon \delta}{j(1-\delta) \log(s/\varepsilon) \log s}.$$

Since $\mathrm{NS}_\delta(f, T^\circ) \geq 0$ for all $T^\circ$, we can bound the number of times we fall into Case 1 by the smallest $t \in \mathbb{N}$ that satisfies

$$\sum_{j=1}^{t} \frac{\varepsilon\delta}{j(1-\delta)\log(s/\varepsilon)\log s} \geq \kappa.$$

Since

$$\sum_{j=1}^{t} \frac{\varepsilon\delta}{j(1-\delta)\log(s/\varepsilon)\log s} \geq \frac{\varepsilon\delta\log t}{(1-\delta)\log(s/\varepsilon)\log s},$$

we conclude that $t = s^{\Theta((\kappa\log(s/\varepsilon)\log s)/\varepsilon\delta)}$ suffices.

Finally, we bound the running time of BUILDSTABILIZINGDT. We first note that Equation (6) implies the follow bounding on the depth $\Delta_j$ of $T_j^\circ$:

$$2^{\Delta_j} \leq \frac{j\log(s/\varepsilon)\log s}{\varepsilon}.$$

Therefore, the score of each of the $j$ leaves of $T_j^\circ$ can be computed in time

$$\mathrm{poly}(2^{\Delta_j}, n^d) = \mathrm{poly}(j, \log s, n^{\log(1/\varepsilon)/\delta}),$$

and so the overall running time of BUILDSTABILIZINGDT is $\mathrm{poly}(t, n^{\log(1/\varepsilon)/\delta})$. This completes the proof of Theorem 4.

## 5 Conclusion and discussion

We have designed a new top-down decision tree learning algorithm that achieves provable guarantees for all target functions $f : \{\pm1\}^n \to \{\pm1\}$ with respect to the uniform distribution. This circumvents impossibility results showing that such a universal guarantee is provably unachievable by existing top-down heuristics such as ID3, C4.5, and CART (and indeed, by any impurity-based heuristic). The analysis of our algorithm draws on Fourier-analytic techniques, and is based on a view of the top-down decision tree construction process as a noise stabilization procedure.

A natural question that our work leaves open is whether our error guarantee can be improved from the current $O(\mathrm{opt}_s) + \varepsilon$ to $\mathrm{opt}_s + \varepsilon$. It would also be interesting to generalize our algorithm and analysis to handle categorical and real-valued attributes, as well as more general classes of distributions. In addition to these technical next steps, there are several broader avenues for future work:

1. *Polynomial-size guarantees for subclasses of target functions?* The focus of our work has been on achieving a universal guarantee: our new algorithm constructs, for all target functions $f$ and size parameters $s$, a decision tree hypothesis of $\mathrm{quasipoly}(s)$ size that achieves error close to that of the best size-$s$ decision tree $f$. It would be interesting to develop more fine-grained guarantees on the performance of our algorithm. Are there broad and natural subclasses of target functions for which our $\mathrm{quasipoly}(s)$ bound on the size of the decision tree hypothesis can be improved to $\mathrm{poly}(s)$, or even $O(s)$?

2. *Harsher noise models?* We have shown that our algorithm works in the agnostic setting—that is, it retains its guarantees even if the samples are corrupted with adversarial label noise. Could our algorithm, or extensions of it, withstand even harsher noise models such as malicious [Val85, KL93] or nasty noise [BEK02]?

3. *Random forests and gradient boosted decision trees.* Looking beyond top-down algorithms for learning a single decision tree, we are hopeful that our work will also lead to an improved theoretical understanding of successful ensemble methods such as random forests and gradient boosted decision trees. Could our techniques be combined with the ideas underlying these methods to give an algorithm that constructs an ensemble of $\mathrm{poly}(s)$ many decision trees of $\mathrm{poly}(s)$ size, that together constitute a good hypothesis for $f$?

## Broader Impact

This work does not present any foreseeable societal consequence.

## Acknowledgments and Disclosure of Funding

Guy, Jane and Li-Yang were supported by NSF grant CCF-192179 and NSF CAREER award CCF-1942123. Neha was supported by NSF award 1704417 and Moses Charikar's Simons Investigator grant.

## Footnotes

[1]Proposition 41 of [BLT20b] is stated for monotone functions $f$, but its proof does not rely on the monotonicity of $f$.

[2]The notion of "noise stability" we referred to in our introduction is defined to be $1 - 2\,\mathrm{NS}_\delta(f)$, though we will work exclusively with noise sensitivity throughout this paper.

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
