[Supplementary Material]

## A  Other related work

Kearns and Mansour [KM99] (see also [Kea96]) analyzed top-down impurity-based heuristics from the perspective of *boosting*, where the attributes queried in the tree are viewed as weak hypotheses.

Recent work of Blanc et al. [BLT20b] gives a top-down algorithm for learning decision trees that achieves provable guarantees for all target functions $f$. However, their algorithm makes crucial use of *membership queries*, which significantly limits its practical applicability and relevance. Furthermore, their guarantees only hold in the realizable setting, requiring that $f$ is itself a size-$s$ decision tree (i.e. $\mathrm{opt}_s = 0$).

There has been extensive work in the learning theory literature on learning the concept class of decision trees [EH89, Blu92, KM93, OS07, GKK08, HKY18, CM19]. However, none of these algorithms proceed in a top-down manner like the practical heuristics that are the focus of this work; indeed, with the exception [EH89], these algorithms do not return a decision tree as their hypothesis. ([EH89]'s algorithm constructs its decision tree hypothesis in a *bottom-up* manner.)

## B  Proof of Fact 2.1

Fact 2.1 is a simple consequence of the following lemma, whose proof also appears in [Jon16]:

**Lemma B.1.** *For all $f : \{\pm 1\}^n \to \{\pm 1\}$ and $i \in [n]$,*

$$\mathrm{NS}_\delta(f) = \tfrac{1}{2}\mathrm{NS}_\delta(f_{x_i=-1}) + \tfrac{1}{2}\mathrm{NS}_\delta(f_{x_i=1}) + \tfrac{\delta}{2(1-\delta)}\cdot \mathrm{Inf}_i^{(\delta)}(f).$$

*Proof.* Let $\boldsymbol{x} \sim \{\pm 1\}^n$ be uniform random, and $\tilde{\boldsymbol{x}} \sim_\delta \boldsymbol{x}$ be a $\delta$-noisy copy of $\boldsymbol{x}$. We first note that

$$
\begin{aligned}
\mathbb{E}[f(\boldsymbol{x})f(\tilde{\boldsymbol{x}})] &= \Pr[\boldsymbol{x}_i = \tilde{\boldsymbol{x}}_i]\cdot \mathbb{E}[f(\boldsymbol{x})f(\tilde{\boldsymbol{x}}) \mid \boldsymbol{x}_i = \tilde{\boldsymbol{x}}_i] + \Pr[\boldsymbol{x}_i \neq \tilde{\boldsymbol{x}}_i]\cdot \mathbb{E}[f(\boldsymbol{x})f(\tilde{\boldsymbol{x}}) \mid \boldsymbol{x}_i \neq \tilde{\boldsymbol{x}}_i] \\
&= \left(1 - \tfrac{\delta}{2}\right)\left(\tfrac{1}{2}\mathbb{E}[f(\boldsymbol{x}^{i=1})f(\tilde{\boldsymbol{x}}^{i=1})] + \tfrac{1}{2}\mathbb{E}[f(\boldsymbol{x}^{i=-1})f(\tilde{\boldsymbol{x}}^{i=-1})]\right) \\
&\quad + \tfrac{\delta}{2}\left(\tfrac{1}{2}\mathbb{E}[f(\boldsymbol{x}^{i=1})f(\tilde{\boldsymbol{x}}^{i=-1})] + \tfrac{1}{2}\mathbb{E}[f(\boldsymbol{x}^{i=-1})f(\tilde{\boldsymbol{x}}^{i=1})]\right).
\end{aligned}
\tag{7}
$$

Next, we have that

$$
\begin{aligned}
\mathbb{E}[D_i f(\boldsymbol{x})D_i f(\tilde{\boldsymbol{x}})] &= \tfrac{1}{4}\mathbb{E}\left[(f(\boldsymbol{x}^{i=1}) - f(\boldsymbol{x}^{i=-1}))(f(\tilde{\boldsymbol{x}}^{i=1}) - f(\tilde{\boldsymbol{x}}^{i=-1}))\right] \\
&= \tfrac{1}{4}\mathbb{E}[f(\boldsymbol{x}^{i=1})f(\tilde{\boldsymbol{x}}^{i=1})] + \tfrac{1}{4}\mathbb{E}[f(\boldsymbol{x}^{i=-1})f(\tilde{\boldsymbol{x}}^{i=-1})] \\
&\quad - \tfrac{1}{4}\mathbb{E}[f(\boldsymbol{x}^{i=1})f(\tilde{\boldsymbol{x}}^{i=-1})] - \tfrac{1}{4}\mathbb{E}[f(\boldsymbol{x}^{i=-1})f(\tilde{\boldsymbol{x}}^{i=1})].
\end{aligned}
\tag{8}
$$

Combining Equations (7) and (8),

$$
\begin{aligned}
\mathbb{E}[f(\boldsymbol{x})f(\tilde{\boldsymbol{x}})] &= \tfrac{1}{2}\mathbb{E}[f(\boldsymbol{x}^{i=1})f(\tilde{\boldsymbol{x}}^{i=1})] + \tfrac{1}{2}\mathbb{E}[f(\boldsymbol{x}^{i=-1})f(\tilde{\boldsymbol{x}}^{i=-1})] - \delta\,\mathbb{E}[D_i f(\boldsymbol{x})D_i f(\tilde{\boldsymbol{x}})] \\
&= \tfrac{1}{2}\mathbb{E}[f_{x_i=1}(\boldsymbol{x})f_{x_i=1}(\tilde{\boldsymbol{x}})] + \tfrac{1}{2}\mathbb{E}[f_{x_i=-1}(\boldsymbol{x})f_{x_i=-1}(\tilde{\boldsymbol{x}})] - \tfrac{\delta}{1-\delta}\cdot \mathrm{Inf}_i^{(\delta)}(f).
\end{aligned}
$$

Since $\mathrm{NS}_\delta(f) = \Pr[f(\boldsymbol{x}) \neq f(\tilde{\boldsymbol{x}})] = \tfrac{1}{2} - \tfrac{1}{2}\mathbb{E}[f(\boldsymbol{x})f(\tilde{\boldsymbol{x}})]$, the lemma follows from the above by rearranging. □

*Proof of Fact 2.1.* We first note that

$$
\begin{aligned}
\mathrm{NS}_\delta(f, T^\circ_{\ell^\star \to x_i}) &= \sum_{\text{leaves } \ell\, \in\, T^\circ_{\ell^\star \to x_i}} 2^{-|\ell|}\cdot \mathrm{NS}_\delta(f_\ell) \\
&= \sum_{\text{leaves } \ell\, \in\, T^\circ} 2^{-|\ell|}\cdot \mathrm{NS}_\delta(f_\ell) \\
&\quad + 2^{-(|\ell^\star|+1)}\cdot \mathrm{NS}_\delta(f_{\ell^\star,x_i=-1}) + 2^{-(|\ell^\star|+1)}\cdot \mathrm{NS}_\delta(f_{\ell^\star,x_i=1}) - 2^{-|\ell^\star|}\cdot \mathrm{NS}_\delta(f_{\ell^\star}) \\
&= \mathrm{NS}_\delta(f, T^\circ) + 2^{-|\ell^\star|}\left(\tfrac{1}{2}\mathrm{NS}_\delta(f_{\ell^\star,x_i=-1}) + \tfrac{1}{2}\mathrm{NS}_\delta(f_{\ell^\star,x_i=1}) - \mathrm{NS}_\delta(f_{\ell^\star})\right).
\end{aligned}
$$

Applying Lemma B.1 with its '$f$' being $f_{\ell^\star}$, we have that

$$\tfrac{1}{2}\mathrm{NS}_\delta(f_{\ell^\star,x_i=-1}) + \tfrac{1}{2}\mathrm{NS}_\delta(f_{\ell^\star,x_i=1}) - \mathrm{NS}_\delta(f_{\ell^\star}) = -\tfrac{\delta}{2(1-\delta)}\cdot \mathrm{Inf}_i^{(\delta)}(f_{\ell^\star}),$$

and this completes the proof. □

## C   Proof of Theorem 3

*Proof.* We apply Theorem 2 with '$T$' being $T^\star$, '$g$' being $f_\delta^{\leq d}$, and $\rho$ being the semimetric $\rho(a, b) = (a - b)^2/2$. As shown by [OSSS05] (and as can be easily verified), $\mathrm{Def}_k(\rho) \leq k$ for this choice of $\rho$, and so

$$\mathrm{Cov}_\rho(T^\star, f_\delta^{\leq d}) \leq k \sum_{i=1}^{n} \lambda_i(T^\star) \cdot \tfrac{1}{2} \, \mathbb{E}\left[(f_\delta^{\leq d}(\boldsymbol{x}) - f_\delta^{\leq d}(\boldsymbol{x}^{\sim i}))^2\right]. \tag{9}$$

We first analyze the quantity on the LHS of Equation (9). For $\boldsymbol{x}, \boldsymbol{x}' \sim \{\pm 1\}^n$ uniform and independent,

$$\begin{aligned}
\mathrm{Cov}_\rho(T^\star, f_\delta^{\leq d}) &= \tfrac{1}{2}\big(\mathbb{E}\left[(T^\star(\boldsymbol{x}) - f_\delta^{\leq d}(\boldsymbol{x}'))^2\right] - \mathbb{E}\left[(T^\star(\boldsymbol{x}) - f_\delta^{\leq d}(\boldsymbol{x}))^2\right]\big) \\
&\geq \tfrac{1}{4}\,\mathbb{E}\left[(f_\delta^{\leq d}(\boldsymbol{x}) - f_\delta^{\leq d}(\boldsymbol{x}'))^2\right] - \mathbb{E}\left[(T^\star(\boldsymbol{x}) - f_\delta^{\leq d}(\boldsymbol{x}))^2\right] \\
&= \tfrac{1}{2}\,\mathrm{Var}(f_\delta^{\leq d}) - \mathbb{E}\left[(T^\star(\boldsymbol{x}) - f_\delta^{\leq d}(\boldsymbol{x}))^2\right],
\end{aligned} \tag{10}$$

where the inequality uses the "almost-triangle" inequality $(a - c)^2 \leq 2((a - b)^2 + (b - c)^2)$ for $a, b, c \in \mathbb{R}$. Furthermore, we have

$$\begin{aligned}
\mathrm{Var}(f_\delta) &= \sum_{S \neq \emptyset} (1 - \delta)^{2|S|} \widehat{f}(S)^2 && \text{(Fourier formulas for } f_\delta \text{ (5) and variance (4))} \\
&= \sum_{\substack{S \neq \emptyset \\ |S| \leq d}} (1 - \delta)^{2|S|} \widehat{f}(S)^2 + \sum_{|S| > d} (1 - \delta)^{2|S|} \widehat{f}(S)^2 \\
&\leq \mathrm{Var}(f_\delta^{\leq d}) + \sum_{|S| > d} e^{(-\delta)2|S|} \widehat{f}(S)^2 && (1 + a \leq e^a) \\
&\leq \mathrm{Var}(f_\delta^{\leq d}) + e^{-2d\delta} \sum_{|S| > d} \widehat{f}(S)^2 && (\text{Since } |S| > d) \\
&\leq \mathrm{Var}(f_\delta^{\leq d}) + e^{-2d\delta} && \text{(Parseval's identity (3): } \sum_{S \subseteq [n]} \widehat{f}(S)^2 = 1) \\
&\leq \mathrm{Var}(f_\delta^{\leq d}) + \varepsilon. && (\text{Since } d = \log(1/\varepsilon)/\delta)
\end{aligned}$$

Similarly,

$$\mathbb{E}\left[(T^\star(\boldsymbol{x}) - f_\delta^{\leq d}(\boldsymbol{x}))^2\right] \leq 2\big(\mathbb{E}[(T^\star(\boldsymbol{x}) - f_\delta(\boldsymbol{x}))^2] + \mathbb{E}\left[(f_\delta(\boldsymbol{x}) - f_\delta^{\leq d}(\boldsymbol{x}))^2\right]\big)$$

$$\text{("almost-triangle" inequality)}$$

$$= 2\left(\mathbb{E}[(T^\star(\boldsymbol{x}) - f_\delta(\boldsymbol{x}))^2] + \sum_{|S| > d} (1 - \delta)^{|S|} \widehat{f}(S)^2\right)$$

$$\leq 2\big(\mathbb{E}[(T^\star(\boldsymbol{x}) - f_\delta(\boldsymbol{x}))^2] + \varepsilon\big). \qquad (\text{Since } d = \log(1/\varepsilon)/\delta)$$

Combining these bounds with Equation (10), we have the following lower bound on the LHS of Equation (9):

$$\begin{aligned}
\mathrm{Cov}(T^\star, f_\delta^{\leq d}) &\geq \tfrac{1}{2}(\mathrm{Var}(f_\delta) - \varepsilon) - \big(2\,\mathbb{E}[(T^\star(\boldsymbol{x}) - f_\delta(\boldsymbol{x}))^2] + 2\varepsilon\big). \\
&= \tfrac{1}{2}\,\mathrm{Var}(f_\delta) - \big(2\,\mathbb{E}[(T^\star(\boldsymbol{x}) - f_\delta(\boldsymbol{x}))^2] + \tfrac{5}{2}\varepsilon\big). \tag{11}
\end{aligned}$$

We now turn to analyzing the RHS of Equation (9):

$$k \sum_{i=1}^{n} \lambda_i(T^\star) \cdot \tfrac{1}{2} \, \mathbb{E}\left[(f_\delta^{\leq d}(\boldsymbol{x}) - f_\delta^{\leq d}(\boldsymbol{x}^{\sim i}))^2\right]$$

$$= k \sum_{i=1}^{n} \lambda_i(T^\star) \cdot \tfrac{1}{4} \, \mathbb{E}\left[(f_\delta^{\leq d}(\boldsymbol{x}) - f_\delta^{\leq d}(\boldsymbol{x}^{\oplus i}))^2\right] \qquad (\boldsymbol{x}^{\oplus i} = \boldsymbol{x} \text{ with its } i\text{-th coordinate flipped})$$

$$= k \sum_{i=1}^{n} \lambda_i(T^\star) \cdot \mathbb{E}\left[D_i f_\delta^{\leq d}(\boldsymbol{x})^2\right]$$

$$= k \sum_{i=1}^{n} \lambda_i(T^\star) \cdot \mathrm{Inf}_i(f_\delta^{\leq d}) \qquad\qquad\qquad\qquad\qquad\qquad \text{(Definition 2)}$$

$$= k \cdot \max_{i \in [n]}\left\{\mathrm{Inf}_i(f_\delta^{\leq d})\right\} \cdot \sum_{i=1}^{n} \lambda_i(T^\star) \ \leq \ k \cdot \max_{i \in [n]}\left\{\mathrm{Inf}_i(f_\delta^{\leq d})\right\} \cdot \log s, \qquad (12)$$

where the final inequality holds because

$$\sum_{i=1}^{n} \lambda_i(T^\star) = \sum_{i=1}^{n} \Pr[\, T^\star \text{ queries } \boldsymbol{x}_i \,] = \sum_{\text{leaves } \ell \in T^\star} 2^{-|\ell|} \cdot |\ell| \leq \log s.$$

Finally, we note that:

$$\mathrm{Inf}_i(f_\delta^{\leq d}) = \sum_{\substack{S \ni i \\ |S| \leq d}} (1-\delta)^{2|S|} \widehat{f}(S)^2 \qquad \text{(Fourier formula for influence; Definition 2)}$$

$$\leq \sum_{\substack{S \ni i \\ |S| \leq d}} (1-\delta)^{|S|} \widehat{f}(S)^2 \ = \ \mathrm{Inf}_i^{(\delta,d)}(f).$$

Combining this with Equations (9), (11) and (12) and rearranging completes the proof. $\qquad\qquad\square$

## D   Proofs of Facts 4.1 and 4.2 and Propositions E.1 and E.2

*Proof of Fact 4.1.* This follows by combining the bounds $\mathrm{Inf}(T) \leq \log s$ (see e.g. [OS07]) and $\mathrm{NS}_\delta(f) \leq \delta \cdot \mathrm{Inf}(f)$ for all $f : \{\pm 1\}^n \to \{\pm 1\}$ [O'D14, Exercise 2.42]. $\qquad\qquad\square$

*Proof of Fact 4.2.* Let $\boldsymbol{x} \sim \{\pm 1\}^n$ be uniform random and $\tilde{\boldsymbol{x}} \sim_\delta \boldsymbol{x}$ be a $\delta$-noisy copy of $\boldsymbol{x}$. Then

$$\mathrm{NS}_\delta(f) = \Pr[f(\boldsymbol{x}) \neq f(\tilde{\boldsymbol{x}})]$$
$$\leq \Pr[T(\boldsymbol{x}) \neq T(\tilde{\boldsymbol{x}})] + \Pr[T(\boldsymbol{x}) \neq f(\boldsymbol{x})] + \Pr[T(\tilde{\boldsymbol{x}}) \neq f(\tilde{\boldsymbol{x}})]$$
$$\leq \mathrm{NS}_\delta(T) + 2\Pr[T(\boldsymbol{x}) \neq f(\boldsymbol{x})],$$

where the final inequality uses that fact that $\boldsymbol{x}$ and $\tilde{\boldsymbol{x}}$ are distributed identically. $\qquad\qquad\square$

## E   The case analysis in the proof of Theorem 4

**Case 1:** $\mathbb{E}_{\boldsymbol{\ell}}[\mathrm{Var}((f_\ell)_\delta)] \geq 4 \, \mathbb{E}_{\boldsymbol{\ell}}\left[\|(f_{\boldsymbol{\ell}})_\delta - T_{\mathsf{opt}}^{\mathsf{trunc}}\|_2^2\right] + 7\varepsilon.$

In this case we claim that there is a leaf $\ell^\star$ of $T^\circ$ with a high score, where we recall that the score of a leaf $\ell$ is defined to be

$$\mathrm{score}(\ell) := 2^{-|\ell|} \cdot \max_{i \in [n]}\left\{\mathrm{Inf}_i^{(\delta,d)}(f_\ell)\right\}.$$

Applying Theorem 3 with its '$T^\star$' being $T_{\mathsf{opt}}^{\mathsf{trunc}}$ and its '$f$' being $f_\ell$ for each leaf $\ell \in T^\circ$, we have that

$$\mathop{\mathbb{E}}_{\boldsymbol{\ell}}\left[\max_{i\in[n]}\left\{\mathrm{Inf}_i^{(\delta,d)}(f_{\boldsymbol{\ell}})\right\}\right] \geq \frac{\frac{1}{2}\mathop{\mathbb{E}}_{\boldsymbol{\ell}}[\mathrm{Var}(f_{\boldsymbol{\ell}})_\delta)] - \left(2\mathop{\mathbb{E}}_{\boldsymbol{\ell}}\left[\|T_{\mathsf{opt}}^{\mathsf{trunc}} - (f_{\boldsymbol{\ell}})_\delta\|_2^2\right] + \frac{5}{2}\varepsilon\right)}{\log(s/\varepsilon)\log s} \qquad \text{(Theorem 3)}$$

$$\geq \frac{\varepsilon}{\log(s/\varepsilon)\log s}, \qquad\qquad (13)$$

where the second inequality is by the assumption that we are in Case 1. Equivalently,

$$\sum_{\ell\in T^\circ} 2^{-|\ell|}\cdot\max_{i\in[n]}\left\{\mathrm{Inf}_i^{\delta,d}(f_\ell)\right\} \geq \frac{\varepsilon}{\log(s/\varepsilon)\log s},$$

and so there must exist a leaf $\ell^\star \in T^\circ$ such that

$$\mathrm{score}(\ell^\star) = 2^{-|\ell^\star|}\cdot\max_{i\in[n]}\left\{\mathrm{Inf}_i^{(\delta,d)}(f_{\ell^\star})\right\} \geq \frac{\varepsilon}{|T^\circ|\log(s/\varepsilon)\log s},$$

where $|T^\circ|$ denotes the size of $T^\circ$.

**Case 2:** $\mathop{\mathbb{E}}_{\boldsymbol{\ell}}[\mathrm{Var}((f_\ell)_\delta)] < 4\mathop{\mathbb{E}}_{\boldsymbol{\ell}}\left[\|(f_{\boldsymbol{\ell}})_\delta - T_{\mathsf{opt}}^{\mathsf{trunc}}\|_2^2\right] + 7\varepsilon.$

In this case, we claim that $\mathrm{error}_f(T_f^\circ) \leq O(\mathsf{opt}_s + \kappa + \varepsilon)$. We will need a couple of propositions:

**Proposition E.1.** $\mathop{\mathbb{E}}_{\boldsymbol{\ell}}[\|(f_{\boldsymbol{\ell}})_\delta - f_{\boldsymbol{\ell}}\|_2^2] \leq 4\kappa.$

*Proof.* We first note that

$$\mathop{\mathbb{E}}_{\boldsymbol{\ell}}\left[\|(f_{\boldsymbol{\ell}})_\delta - f_{\boldsymbol{\ell}}\|_2^2\right] \leq 2\mathop{\mathbb{E}}_{\boldsymbol{\ell}}\left[\|(f_{\boldsymbol{\ell}})_\delta - f_{\boldsymbol{\ell}}\|_1\right] \qquad \text{(Since } f_\ell \text{ and } (f_\ell)_\delta \text{ are } [-1,1]\text{-valued)}$$

$$= 2\mathop{\mathbb{E}}_{\boldsymbol{\ell}}\left[\mathop{\mathbb{E}}_{\boldsymbol{x}}\left[|(f_{\boldsymbol{\ell}})_\delta(\boldsymbol{x}) - f_\ell(\boldsymbol{x})|\right]\right]$$

$$= 2\mathop{\mathbb{E}}_{\boldsymbol{\ell}}\left[\mathop{\mathbb{E}}_{\substack{\boldsymbol{x}\\ \tilde{\boldsymbol{x}}\sim_\delta \boldsymbol{x}}}\left[|(f_{\boldsymbol{\ell}})(\tilde{\boldsymbol{x}}) - f_\ell(\boldsymbol{x})|\right]\right]$$

$$= 2\mathop{\mathbb{E}}_{\boldsymbol{\ell}}\left[2\mathop{\Pr}_{\substack{\boldsymbol{x}\\ \tilde{\boldsymbol{x}}\sim_\delta \boldsymbol{x}}}\left[f_{\boldsymbol{\ell}}(\tilde{\boldsymbol{x}}) \neq f_{\boldsymbol{\ell}}(\boldsymbol{x})\right]\right]$$

$$= 4\mathop{\mathbb{E}}_{\boldsymbol{\ell}}\left[\mathrm{NS}_\delta(f_{\boldsymbol{\ell}})\right] = 4\,\mathrm{NS}_\delta(f, T^\circ).$$

By Fact 2.1, we have that $\mathrm{NS}_\delta(f, T^\circ) \leq \mathrm{NS}_\delta(f)$, and the claim follows. $\qquad\square$

**Proposition E.2.** *For each leaf $\ell \in T^\circ$, we have $\mathbb{E}\left[(f_\ell(\boldsymbol{x}) - \mathrm{sign}(\mathbb{E}[f_\ell]))^2\right] \leq 2\mathbb{E}[(f_\ell(\boldsymbol{x}) - c)^2]$ for all constants $c \in \mathbb{R}$.*

*Proof.* Let $p := \Pr[f_\ell(\boldsymbol{x}) = 1]$ and assume without loss of generality that $p \geq \frac{1}{2}$. On one hand, we have that $\mathbb{E}\left[(f_\ell(\boldsymbol{x}) - \mathrm{sign}(\mathbb{E}[f_\ell]))^2\right] = \mathbb{E}\left[(f_\ell(\boldsymbol{x}) - 1)^2\right] = 4(1-p)$. On the other hand, since

$$\mathbb{E}[(f_\ell(\boldsymbol{x}) - c)^2] = p(1-c)^2 + (1-p)(1+c)^2$$

this quantity is minimized for $c = 2p - 1$ and attains value $4p(1-p)$ at this minimum. Therefore indeed

$$\min_c\left\{\mathbb{E}[(f_\ell(\boldsymbol{x}) - c)^2]\right\} = 4p(1-p) \geq 2(1-p) = \frac{1}{2}\mathbb{E}\left[(f_\ell(\boldsymbol{x}) - \mathrm{sign}(\mathbb{E}[f_\ell]))^2\right]$$

and the proposition follows. $\qquad\square$

With Propositions E.1 and E.2 in hand, we are ready to bound $\mathrm{error}_f(T_f^\circ)$. Recall that $T_f^\circ$ is the completion of $T^\circ$ that we obtain by labeling each leaf $\ell$ with $\mathrm{sign}(\mathbb{E}[f_\ell])$. Therefore,

$$\mathrm{error}_f(T_f^\circ) = \mathop{\mathbb{E}}_\ell \left[ \mathrm{dist}(f_\ell, \mathrm{sign}(\mathbb{E}[f_\ell])) \right]$$

$$= \frac{1}{4} \mathop{\mathbb{E}}_\ell \left[ \| f_\ell - \mathrm{sign}(\mathbb{E}[f_\ell]) \|_2^2 \right]$$

$$\leq \frac{1}{2} \mathop{\mathbb{E}}_\ell \left[ \| f_\ell - \mathbb{E}[(f_\ell)_\delta] \|_2^2 \right] \qquad \text{(Proposition E.2)}$$

$$\leq \mathop{\mathbb{E}}_\ell \left[ \| f_\ell - (f_\ell)_\delta \|_2^2 \right] + \mathop{\mathbb{E}}_\ell \left[ \| (f_\ell)_\delta - \mathbb{E}[(f_\ell)_\delta] \|_2^2 \right]$$

$$\leq 4\kappa + \mathop{\mathbb{E}}_\ell [\mathrm{Var}((f_\ell)_\delta)] \qquad \text{(Proposition E.1)}$$

By the assumption that we are in Case 2, we have that:

$$\mathop{\mathbb{E}}_\ell [\mathrm{Var}((f_\ell)_\delta)] < 4 \mathop{\mathbb{E}}_\ell \left[ \| (f_\ell)_\delta - T_{\mathsf{opt}}^{\mathrm{trunc}} \|_2^2 \right] + 7\varepsilon$$

$$\leq 8 \left( 4\kappa + \mathop{\mathbb{E}}_\ell \left[ \| f_\ell - T_{\mathsf{opt}}^{\mathrm{trunc}} \|_2^2 \right] \right) + 7\varepsilon \qquad \text{(Proposition E.1)}$$

$$= 8 \left( 4\kappa + 4 \mathop{\mathbb{E}}_\ell \left[ \mathrm{dist}(f_\ell, T_{\mathsf{opt}}^{\mathrm{trunc}}) \right] \right) + 7\varepsilon$$

$$= 8 \left( 4\kappa + 4(\mathsf{opt}_s + \varepsilon) \right) + 7\varepsilon$$

$$\leq O(\mathsf{opt}_s + \kappa + \varepsilon).$$

and so we have shown that $\mathrm{error}_f(T_f^\circ) \leq O(\mathsf{opt}_s + \kappa + \varepsilon)$.