[Reviews · NeurIPS 2020]

Review 1

Summary and Contributions: The paper proves that it is possible to avoid the impossibility result on universal guarantee on the performance of top-down impurity based decision tree heuristics. The basis of the result is an extension that takes into account small sets of attributes instead of single ones and at each step chooses the expansion leaf and attribute on the basis of increasing noise stability.

Strengths: Showing achievable universal guarantee for arbitrary functions is a significant step in the area. Analogous approaches and result have recently appeared for subclasses of fucntions (e.g., monotone functions [ICML 2020]). The effective exploitation of the conncetion to noise sensitivity is also an interesting strong point.

Weaknesses: The main weakness is the quasipolynomiality of the bounds: size of the tree and complexity. It is not clear whether, due to the approach, this is actually improvable for arbitrary functions. It would be nice to argue at some more length the Remark 1, giving more details about the necessity of the attenuating factor and the existence, in its absence, of situations analogous to the impossibility result.

Correctness: As far as I can see all the claims appear to be sound and, the description of steps not fully included in the body of the paper is clearly managed.

Clarity: The paper appears to be clearly written and also the most involved parts are not difficult to follow.

Relation to Prior Work: The bibliografy and the related works are sufficiently discussed.

Reproducibility: Yes

Additional Feedback:


Review 2

Summary and Contributions: This paper considers the problem of learning decision trees. You are given samples from a function f on the Boolean cube that is known to be computed by a size s decision tree. The goal is to produce a hypothesis h that is also a small decision tree and is close to f. It was known that simply looking at correlations is not a good idea, simple functions like parity of a few variables would defeat this algorithm. Indeed, I don't think there was any known algorithm that was guaranteed to return a "decision tree" of small size. This paper presents an algorithm of this type. - The learnt hypothesis h is a decision tree of size exp((log (s))^3) so it is superpolynomial in s. In fact, if the orgiinal function is not exactly computed by a small decision tree, then the error we get is O(opt) where opt is the error of the best size s decision tree. - The algorithm is based on a simple splitting criterion where you look at something called the low-degree influence of a variable rather than just correlations. - The only downside is that there is no simple way to compute low-degree influence. To compute degree d influences requires n^d time which is why their running time and size are not great.

Strengths: This is a problem that has attracted considerable attention in the Fourier analysis/computational learning community. I would say that this will be regraded as an important result in that community. The result is strong, and the techniques (which rely on an old result by OSSS on influences in decision trees) are elegant.

Weaknesses: This is a purely theoretical result for a number of reasons: the model (learning a Boolean function from uniform examples), the problem statement, the running time. The advantage of top-down splitting rules is simplicity. The rules the authors suggest, which involve computing low degree influences are arguably not so simple. Well perhaps they are simple to state, but not to implement. In fairness, the authors do not claim otherwise. This paper fits well in a long line of work in theory (starting from Linial, Mansour, Nisan) on learning under the uniform distribution using Fourier analysis tools. A more theoretical venue (like say COLT) would expose this paper to an audience that can better appreciate the intricate Fourier analysis used in the proof. (But having said that, the authors chose to send it to NeuRIPS and it is clearly above the bar in my view).

Correctness: Yes, though I did not check the proof in detail.

Clarity: Very.

Relation to Prior Work: Yes.

Reproducibility: Yes

Additional Feedback:


Review 3

Summary and Contributions: The paper gives an decision tree learning algorithm in the agnostic setting with uniform distribution on the input. Both the domain and the range are binary. The running time and the size of the decision tree are super-polynomial. The paper employs techniques from Boolean analysis of functions.

Strengths: This appears to be the first work to give a decision tree learning algorithm with provable guarantees in a general agnostic setting. Their splitting criterion seems novel and is interesting.

Weaknesses: The paper is mainly of theoretical interest only as far as I can tell. The paper does not consider the question of practical application, even though in the motivation comparison with practical algorithms are invoked. Certainly, the proof technique would be hard to carry over to more general setting, but I wonder if the algorithm itself leads to interesting practical results. For that one would need to generalize the splitting criterion to these settings. Impracticality arises in part because of the complexity of the algorithm which is higher than those used in practice; but another important limitation appears to be restriction to binary domain with uniform inputs. While one may be able to extend it to other product distribution, it's not clear how to extend it to non-binary domains and non-product distributions. In a way, the hard examples cited at the beginning are not relevant in practice as they are unlikely to arise.

Correctness: I only read the main part of the paper which has the proof sketch. It appears correct.

Clarity: Well written.

Relation to Prior Work: Well done.

Reproducibility: Yes

Additional Feedback: While \delta is taken as a parameter by the algorithm, it doesn't appear in the theorem statement. It's instantiated later in the proof. It would be useful to also specify it in the theorem statement.

[Author Response · NeurIPS 2020]

We thank the reviewers for their thoughtful and valuable feedback. We appreciate their time and effort, especially given the current uncertain times.

There are no factual inaccuracies in the reviews that we would like to correct.

We agree with the reviewers that the contributions of this paper are mostly theoretical in nature. However, the class of algorithms that we study, top-down induction of decision trees, is of significant practical relevance, and we believe that it is interesting and important to put these algorithms on firm theoretical footing. The decision tree heuristics ID3, C4.5, and CART (and related ensemble methods) all enjoy empirical success in machine learning practice, but there are still relatively few works giving rigorous guarantees on their performance. A natural first step is to analyze these heuristics and their variants within the PAC model and under feature and distributional assumptions.

We are hopeful that our results and techniques point to concrete avenues for future work that will further close the gap between theory and practice. Regarding our feature and distributional assumptions in particular, there are analogues of noise sensitivity for categorical features and general (non-product) distributions, and therefore one could design and analyze natural extensions of our algorithm to such settings. For future work, it would be interesting to extend our provable guarantees to these more general settings, and to experimentally evaluate the performance of such generalizations of our algorithm on practical datasets.

Finally, we thank the reviewers for their specific suggestions for improving the presentation of our paper. We agree with their suggestions and will implement them: we will add more details to Remark 1, and we will also specify the choice of $\delta$ in the statement of Theorem 1.

[Meta-Review · NeurIPS 2020]

The three reviews agree that the paper develops strong theoretical results regarding an important topic. Also the techniques are interesting, and the paper is well written. The main negative aspect in the reviews concerns the practical applicability of the results. Although the authors address this in their reply, the reviewers after discussion are not really convinced about the potential for bridging the gap between theory and practice. Regardless of this, the reviewers are clear in their assessment that the work deserves publications purely on the strength of the theoretical contribution.